# Association of clinical course with thyroid-stimulating immunoglobulin in Graves' ophthalmopathy in Mongolians

**Oyungerel Bayarmunkh**[1,2], **Chimedlkhamsuren Ganbold**[2], **Sima Das**[3], **Uranchimeg Davaatseren**[4], **Nomin-Erdene Minjuurdorj**[5], **Sarantuya Jav**[2]*

**1** Department of Ophthalmology, The State Third Central Hospital, Ulaanbaatar, Mongolia, **2** Department of Molecular biology and Genetics, Mongolian National University of Medical Sciences, Ulaanbaatar, Mongolia, **3** Departments of Oculoplasty, Dr.Shroff's Charity Eye Hospital, New Delhi, India, **4** Department of Ophthalmology, Mongolian National University of Medical Sciences, Ulaanbaatar, Mongolia, **5** Department of Endocrinology, Diabetes Center, The State Third Central Hospital, Ulaanbaatar, Mongolia

* sarantuya.j@mnums.edu.mn

**Data Availability Statement:** all the study data is attached in supporting information file which named "study data sharing".

## Abstract

Graves' ophthalmopathy (GO) is a complex inflammatory condition affecting the orbit and is often associated with Graves' disease (GD). This study aims to determine the levels of thyroid-stimulating immunoglobulin (TSI) and thyroid-stimulating hormone receptor autoantibody (TSHR-ab) in the serum of patients with GO, compare it with the GD, and determine whether there is a correlation with the clinical course of GO. The cross-sectional study included 82 patients with GO, 81 patients with GD, and 75 healthy subjects. The ocular manifestations of GO were identified and evaluated by the clinical activity score (CAS) and severity of GO using the European Group of Graves' Orbitopathy (EUGOGO). TSI and TSHR-ab levels in the serum of participants were determined with ELISA kits and correlated with clinical findings. A total of 238 participant's data were analyzed. There were 14 patients (17%) with unilateral GO. The most common ocular signs were eyelid retraction 68 (82.3%) and proptosis 61 (74.4%). The mean CAS score was 2.65±1.64 in GO patients and was higher in men than women ($P = 0.008$). The mean of TSI was 37.95±35.41 in GO, 14.16±15.67 in GD, and 4.33±2.94 in healthy controls ($P<0.0001$). The TSI was significantly higher in patients with GO than in those with GD ($P<0.0001$). There were no correlations between TSI and TSHR-ab levels and CAS scores. However, we observed a correlation between the TSI level and the severity of GO ($P = 0.023$). The area under the ROC curve (AUC) of TSI was 0.933 and selected 14.1 IU/ml was the optimal cutoff value (98.78% of sensitivity, 83.97% of specificity). Our study showed that TSI is significantly related to GO and the severity of GO. Therefore, TSI can be used as a predictor of severe GO to help in prognostication, follow-up and treatment planning.

## Introduction

Graves' ophthalmopathy (GO) is an autoimmune inflammatory disorder of the orbital tissue, that is generally associated with hyperthyroidism of Graves' disease (GD). It occurs in 25–50%

**Funding:** This study was supported by project funding the Mongolian National University of Medical Sciences (MNUMS). The funders had no role in the study design, data collection, and analysis, decision to publish, or preparation of the manuscript.

**Competing interests:** The authors have declared that no competing interests exist.

of GD patients, mostly in women aged 30–60 [1,2]. Although GO pathogenesis has not been fully elucidated, the production of thyroid-stimulating hormone receptor (TSHR) autoantibody (TSHR-ab) activates TSHR on the orbital fibroblasts and preadipocytes, which causes increased adipose tissue and extraocular muscle enlargement [3]. Typical and frequent clinical signs of GO are proptosis and eyelid retraction [4–6]. In severe cases, symptoms of exposure keratopathy, dysfunction of ocular motility, optic neuropathy, and globe subluxation can impair vision, cosmetic appearance, and quality of life [4,5]. The clinical course of GO varies, but it is usually mild to moderately severe, severe. Approximately 3–5% of patients develop severe sight-threatening forms due to high GO activity and poor control [1]. GO has an inflammatory active phase that lasts for approximately 18 months (range, 3 to 36 months) followed by a stable or inactive phase [7]. Therefore, ophthalmological assessment and long-term follow-up are necessary for patients with GO to prevent complications. Although few thyroid tests are used to diagnose and decide the management of GO, the lack of specific indicators for GO in diagnosis and treatment has been a challenge for clinicians and patients [8].

Previous studies have focused on determining whether TSHR-ab levels are associated with GO. The levels of both blocking and stimulating TSHR autoantibodies in GO have been measured using different laboratory assays, and the thyroid-stimulating autoantibodies (TSI) have been shown to be more associated with the ocular manifestation of GO [7–12]. Although the studies have shown an increase in TSI levels in GOs, there are differences in whether they are related to ocular signs, activity, or severity [7,8,11–14]. SY Jang et al. and Lytton et al. reported that TSI levels were high in GO; its sensitivity/specificity was more specific than TSHR-ab in GO, according to a reporter assay [8,11]. Gabriela et al. showed that TSHR-ab levels in GO were correlated with antithyroid drugs but not related to GO clinical features [14].

Although ophthalmologists can monitor the clinical symptoms of GO, the overall prognosis is unpredictable. There is a need to establish a test that can determine the clinical course of GO. Therefore, we aimed to determine the levels of TSI and TSHR-ab in the serum of patients with GO, compare it with the GD and healthy subjects, and determine whether these levels are predictive of disease severity and activity.

## Materials methods

### Patients

We performed a cross-sectional study in 163 Graves' disease patients and 75 healthy subjects who visited the Department of Ophthalmology and Endocrinology, The State Third Central Hospital, Ulaanbaatar, in 2020. The study was approved by the Ethical Review Board at the Mongolian National University of Medical Sciences in accordance with the principles of the Declaration of Helsinki. Written consent was obtained from all participants. Among the enrolled patients, 82 patients presented with ocular features of GO, and 81 patients presented without ocular symptoms and orbital involvement. Endocrinologists made the GD diagnosis based on thyroid examination and thyroid function tests. A single ophthalmologist performed the clinical ophthalmic examination and diagnosed GO using assessment of history, proptosis (Hertel measurement ≥ 17mm or 2 mm differences between the eyes), eyelid retraction (>1 mm in primary gaze), optic neuropathy, ocular motility, corneal or conjunctival involvement, and orbital computed tomography scan. All GO patients had at least one sign of the disease, e.g., proptosis. The activity of GO was assessed according to seven items of the clinical activity score (CAS) based on spontaneous retrobulbar pain, pain on attempted gaze, redness of the conjunctiva, redness of the eyelid, swelling of the eyelid, chemosis and swelling of the caruncle. One point is given for each item presents and, it ranges from 0–7 scores in total. A score of ≥3 indicates the activity of GO [15,16]. The severity of GO was assessed using the severity

classification of European Group of Graves' Orbitopathy (EUGOGO), which is classified as mild, moderate to severe (lid retraction > 2 mm, proptosis > 3 mm, diplopia, moderate soft tissue involvement), or sight-threatening (optic neuropathy, corneal breakdown) [16,17]. The duration of disease, treatment adherence, follow-up, smoking, diabetes and hypertension were assessed from history. The patients diagnosed with GO, did not have any treatment and orbital surgery for orbitopathy. Participants who did not take antithyroid drugs (ATD) or did not take ATD regularly were defined as having poor treatment adherence. Participants who took the ATD according to prescriptions were defined as having good treatment adherence.

## Laboratory assay

Blood samples were taken from a vein in the morning within a week after the GO examination and immediately sent to the Immunology Laboratory of Mongolian National University of Medical Sciences. The serum was separated by centrifugation and stored at -20˚C. The collected samples were assessed within a month of blood collection using a Sunglong Human Thyroid-stimulating Immunoglobulin (TSI) enzyme-linked immunosorbent assay (ELISA) kit and a human thyroid-stimulating hormone receptor antibody (TSHR-ab) ELISA kit. Based upon the principles of the sandwich ELISA method, the optical density (OD) was measured spectrophotometrically at a wavelength of 450 nm. The concentrations of TSI and TSHR-ab in the samples were calculated by comparing the OD of the samples to the standard curve. According to the manufacturer's instructions, TSI ranges from 3 to 200 IU/ml, and TSHR-ab ranges from 80 to 4000 pg/ml.

## Data analysis and statistics

Statistical analysis was performed by Stata 13.1 software and GraphPad Prism 8.0 software. The general parameters were expressed numerically as the means with a standard deviation (SD) for normally distributed variables, medians with ranges for abnormally distributed variables, or numbers with percentages for categorical variables. The Kruskal-Wallis test was used to calculate the significance of the differences between the three groups. Groups were statistically compared using the chi-square test, the Mann Whitney-U test, or Fisher's exact test. The correlations of serum TSI and TSHR-ab with CAS were calculated via Spearman correlation analysis. Receiver operating characteristic (ROC) curve TSI and TSHR-ab plotted using GO and GD with controls and analyzed to select the best cutoff level. The sensitivity and specificity of both TSI and TSHR-ab were also examined. $P$ values <0.05 were considered statistically significant.

## Results

In this cross-sectional study, we included 82 patients with GO, 81 patients with GD and 75 healthy subjects, and all participants were adults. The demographic and clinical data are shown in Table 1. The mean age of the healthy controls was 47.32±14.22 years, with 61 females (81.3%). The mean age of GO patients was 42.63±11.03, which was younger than that of GD patients ($P$ = 0.012). Four patients with GO and six patients with GD were not treated for GD. GO was present in 25.6% (21) of current smokers, which was larger than the proportion with GD (3.7%, or 3 patients; $P$ = 0.0001). The most common ocular signs were eyelid retraction 68 (82.3%) and proptosis 61 (74.4%). Upper eyelid retraction occurred in 60 patients (73%), which was significantly higher than lower eyelid retraction ($P$<0.0001). There were no differences in proptosis between the right (18 mm, median) and left eyes (17.5 mm, median) ($P$ = 0.49). The mean CAS score was 2.65±1.64 and varied depending on sex and smoking. It was 4.33±1.88 in men, which was statistically higher than women ($P$ = 0.008) (Fig 1), but 3.52±1.7 in smokers, which was higher than non-smokers with no statistically significant differences (P = 0.872).

Table 1. Demographic and clinical data of GO patients.

| Parameters | GO (n = 82) | GD (n = 81) | P value |
|---|---|---|---|
| Age (y, mean±SD) | 42.63 ±11.03 | 47.49±10.87 | 0.012 |
| Female gender | 70 (85.4) | 75 (92.6) | n.s |
| Duration of GD (y, median and range) | 6 (0–35) | 4.5 (0–36) | 0.039 |
| Current smokers | 21 (25.6) | 3 (3.7) | 0.0001 |
| Diabetes mellitus | 10 (12.2) | 6 (7.4) | n.s. |
| Hypertension | 14 (17.1) | 34 (42) | 0.0005 |
| Family history of thyroid disease | 37 (45.1) | 43 (53.1) | n.s. |
| Antithyroid drug | 77 (93.9) | 74 (91.4) | n.s. |
| L-Thyroxine | 4 (4.9) | 5 (6.2) | n.s. |
| Thyroidectomy (total and subtotal) | 10 (12.2) | 7 (8.6) | n.s. |
| Radioiodine treatment | 3 (3.7) | 2 (2.5) | n.s. |
| **Ocular manifestations** | | | |
| Unilateral eyelid retraction | 9 (11) | | |
| Unilateral GO | 14 (17) | | |
| Eyelid retraction | 68 (82.3) | | |
| Upper | 60 (73) | | |
| Lower | 23 (28) | | |
| Proptosis | 61 (74.4) | | |
| Proptosis (mm, median and range) Right eye | 18 (10–29) | | |
| Left eye | 17.5 (11–26) | | |
| Diplopia | 11 (13.4) | | |
| Optic nerve involvement | 5 (6.1) | | |
| Chemosis | 7 (8.5) | | |
| CAS (0–7) | 2.65±1.64 | | |

Notes: The numerical values are reported as the mean ± standard deviation or number (%) with the exception of GD duration and proptosis, which are presented as the median (range).

Abbreviation: GO, Graves' ophthalmopathy; GD, Graves' disease; CAS, clinical activity score; T3, triiodothyronine; fT4, free thyroxine; TSH, thyroid-stimulating hormone.

Both serum TSI and TSHR-ab levels in GD and GO were higher than in healthy controls (Fig 2A and 2B). However, the TSI level in GO was significantly higher than that in GD, approximately 2.7 times higher ($P<0.0001$). The TSI and TSHR-ab levels in the GD and GO data are shown in Table 2. The TSI level was correlated with the severity of GO and was highest in the moderate-severe stage ($P<0.023$). The CAS of GO were not related to TSI and TSHR-ab levels respectively (Spearman's Rho = -0.013, Rho = 0.066, $P = 0.904$, $P = 0.555$). In GD and GO, TSI and TSHR-ab levels were strongly correlated with GD treatment status ($P<0.001$). The ROC curve analysis was used to compare GO and GD patients with healthy subjects. The areas under the ROC curves (AUCs) of TSI and TSHR-ab were 0.933 and 0.908, respectively. We selected a cutoff value of 14.1 IU/ml in TSI (98.78% sensitivity, 83.97% specificity) and 106.04 pg/ml in TSHR-ab (80.98% of sensitivity, 96% of specificity) (Figs 3 and 4).

## Discussion

GO is a complex inflammatory condition affecting the orbit and is often associated with GD [18]. GO has specific ocular clinical features and is relatively easy to diagnose in moderate and

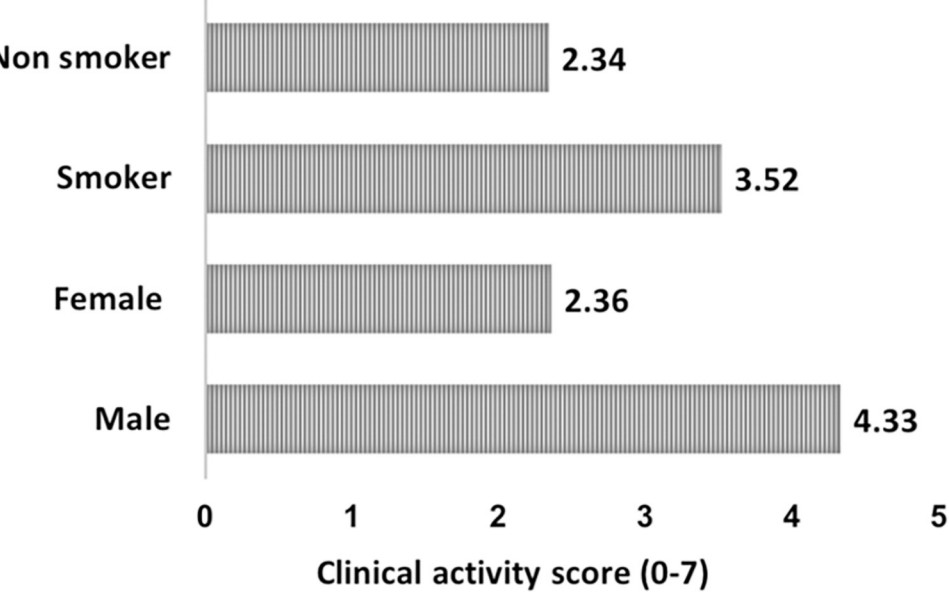

**Fig 1. The mean of clinical activity score (CAS) in different groups with Graves' ophthalmopathy.** The mean CAS score was significantly higher in males than in females ($P = 0.008$). The mean CAS score in smokers with GO was higher than that in non-smokers, with no statistically significant differences ($P = 0.872$).

severe stages [19]. However, sometimes the prognosis is unpredictable. Some orbital inflammatory disease shows similar features of GO which makes it difficult to diagnose GO. The parameters of autoimmune thyroid disease are widely used in GD management, but the lack of indicators for the prognosis and treatment of GO has been a challenge for clinicians and patients [8,19]. The main issue is that the pathogenesis of GO is not fully defined, explained by the production of thyroid-stimulating hormone receptor autoantibodies [3,20,21,22]. The relationship between clinical ocular features and TSH receptor autoantibodies has been studied in many advanced assays [21,23,24]. Some previous studies showed that GO is related to TSI and TSHR-ab levels, which is a good opportunity to compare our results. We determined the TSHR-ab and TSI levels of patients with GD and GO in the study using ELISA kits and examined whether they were associated with ophthalmopathy. ELISA is a widely used method that

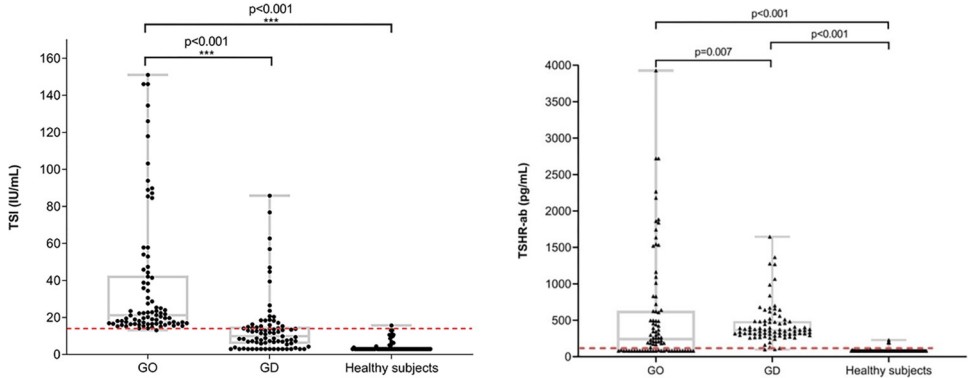

**Fig 2.** a, b. TSI and TSHR-ab levels in each group. Abbreviation: GO, Graves' ophthalmopathy; GD, Graves' disease.

**Table 2. The TSI and TSHR-ab levels in GD and GO.**

| Parameters | | TSI (IU/ml) | TSHR-ab (pg/ml) |
|---|---|---|---|
| Graves' disease (N = 81) | | 14.16±15.67 | 445.24±278.51 |
| Graves' ophthalmopathy (N = 82) | | 37.95±35.41 | 570.87±759.64 |
| Healthy subjects (N = 75) | | 4.33±2.94 | 85.97±24.72 |
| P value | | <0.0001[a] | <0.0001[a] |
| Activity of GO | Inactive stage CAS<3 (N = 56) | 36.68±31.88 | 596.73±763.48 |
| | Active stage CAS≥3 (N = 26) | 40.66±42.61 | 573.32±766.35 |
| | P value | 0.271[b] | 0.582[b] |
| Severity of GO | Mild (N = 29) | 32.77±31.96 | 524.46±671.86 |
| | Moderately severe (N = 48) | 43.3±38.17 | 639.48±838.19 |
| | Sight-threatening (N = 5) | 16.48±1.03 | 181.37±93.39 |
| | P value | 0.023[a] | 0.422[a] |
| ATD treatment adherence in GO | Good (N = 52) | 21.47±18.68 | 260.32±426.94 |
| | Poor or untreated (N = 30) | 66.5±39.47 | 1109.16±904.75 |
| | P value | <0.001[b] | <0.001[b] |
| ATD treatment adherence in GD | Good (N = 64) | 8.57±4.60 | 343.12±108.97 |
| | Poor or untreated (N = 17) | 35.23±23.41 | 829.7±377.26 |
| | P value | <0.001[b] | <0.001[b] |

Notes: The numerical values are reported as the mean ± standard deviation. *P* values calculated by [a]the Kruskal-Wallis test or [b]the Mann-Whitney U test.

Abbreviations: TSI, thyroid-stimulating immunoglobulin; TSHR-ab, thyroid-stimulating hormone receptor antibody; GO, Graves' ophthalmopathy; GD, Graves' disease; CAS, clinical activity score; ATD, antithyroid drug.

is low cost and quick to respond to. We determined TSHR-ab and TSI levels within the same week of evaluating ocular features of GO, which allowed for more realistic results.

Our results show that an increase and decrease in TSI and TSHR-ab levels in patients with GO and GD correlate with treatment status and level of control of the systematic thyroid status. However, in GO, TSI is more significant than TSHR-ab. On long-term posttreatment follow-up in some patients, TSHR ab levels can return to normal in both GD and GO patients, showing no correlation with ocular manifestation. However, the TSI level remained higher in patients with GO than in those with GD (P<0.0001). In this study, the clinical activity score in GO patients was assessed and found to have not correlated with TSI and TSHR-ab levels. However, TSI levels were related to the severity of GO. TSI levels increased with an increase in severity of GO from the mild stage (32.77±31.96 IU/ml) to the moderate to severe stage (43.3 ±38.17 IU/ml) and decreased in the sight-threatening stage (16.48±1.03 IU/ml) (P = 0.023). The activity of the disease is highest in the moderate to severe stage. Even if the active phase subsides, the patients may still be classified in the sight-threatening stage due to difficult irreversible structural changes, such as optic neuropathy and corneal breakdown due to exposure keratopathy. This may also be due to the small number of GO patients involved, which is a limitation of our study. The studies reported that smoking is a risk factor for GO, and our study also found that current smoking in GO patients (25.6%) was higher than that in GD patients (3.7%) (P = 0.001) [25–27].

Gabriela et al. showed that TSHR-ab and TSI levels in untreated GO are positively high but not related to clinical symptoms of GO but can decrease during treatment with corticosteroids [14]. Young et al. showed that TSI levels are not related to CAS scores but are related to NOS-PECS scores and are more related to GO than other antibodies [7]. We have expressed similarities with the results of Young and Gabriela. In contrast, Lytton et al. and SY Jang et al. found

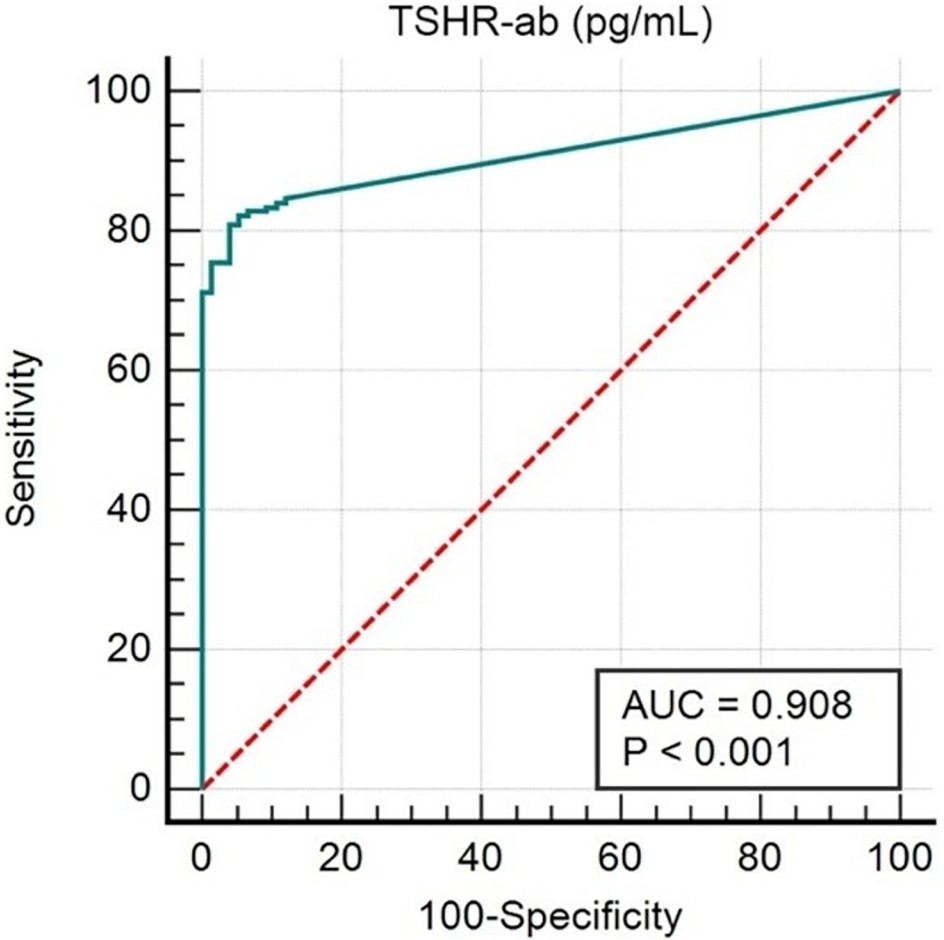

**Fig 3. ROC curve analysis comparing TSHR-ab between GO and GD vs. controls.** Cutoff value >106.04 pg/mL.
Area under the ROC curve (AUC) = 0.908. (95% CI, 0.864–0.942), p<0.0001. Positive percentage agreement (PPA) or.
Sensitivity = 80.98% (95% CI, 74.1–86.7). Negative percentage agreement (NPA) or Specificity = 96.0 (95% CI, 88.8–99.2). Youden Index J = 0.7698.

that the activity of GO was related to TSI levels, which may be due to the sensitivity and clinical feature assessment of laboratory methods [8,11]. The fact that TSI levels in not treated with the antithyroid drug GO are higher than those in GD patients leads to a consensus that the increase in antibodies stimulates orbital receptors. However, TSI and TSHR-ab levels can be gradually reduced with appropriate treatment [14,28]. Although these two parameters return to normal after treatment, there is another hypothesis that the TSI level in GO remains slightly higher than that in GD, which may lead to a gradual increase in ophthalmic clinical features. Thus, we can use TSHR-ab and TSI indicators together in ophthalmic practice and obtain more information from TSI levels to monitor GO. However, in our further studies, we will continue to look for more relevant indicators to use in ophthalmology.

## Conclusions

We showed that TSI is associated with GO. Determining TSI can help in early diagnosis and proper treatment of GO. However, further research is needed to define the pathogenesis and prognosis of GO.

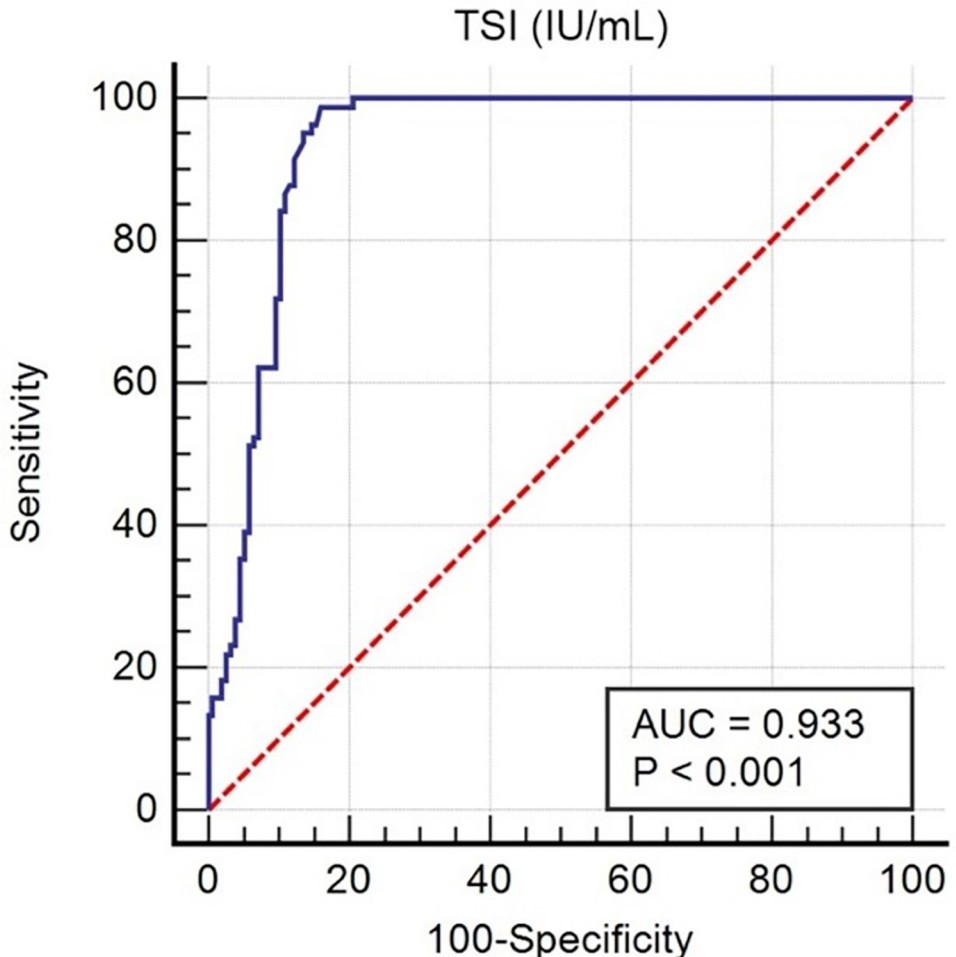

**Fig 4. ROC curve analysis comparing TSI between GO vs. GD and controls.** Cutoff value >14.1 IU/mL Area under the ROC curve (AUC) = 0.933. (95% CI, 0.893–0.961), p<0.0001. Positive percentage agreement (PPA) or Sensitivity = 98.78% (95% CI, 93.4–100.0). Negative percentage agreement (NPA) or Specificity = 83.97% (95% CI, 77.3–89.4). Youden Index J = 0.8275.

## Supporting information

**S1 Data.**
(XLSX)

## Author Contributions

**Conceptualization:** Sima Das, Uranchimeg Davaatseren, Sarantuya Jav.

**Data curation:** Oyungerel Bayarmunkh, Chimedlkhamsuren Ganbold.

**Formal analysis:** Chimedlkhamsuren Ganbold.

**Investigation:** Oyungerel Bayarmunkh, Chimedlkhamsuren Ganbold, Sima Das, Uranchimeg Davaatseren, Nomin-Erdene Minjuurdorj.

**Methodology:** Oyungerel Bayarmunkh, Sima Das, Uranchimeg Davaatseren, Nomin-Erdene Minjuurdorj.

**Project administration:** Sarantuya Jav.

**Supervision:** Sarantuya Jav.

**Validation:** Sarantuya Jav.

**Writing – original draft:** Oyungerel Bayarmunkh.

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
