## [Decision Letter · Decision Letter 0]

1 Jul 2022

PONE-D-22-03512Association of clinical course with thyroid-stimulating immunoglobulin in Graves’ ophthalmopathy in MongoliansPLOS ONE

Dear Dr. Jav,

Thank you for submitting your manuscript to PLOS ONE. After careful consideration, we feel that it has merit but does not fully meet PLOS ONE’s publication criteria as it currently stands. Therefore, we invite you to submit a revised version of the manuscript that addresses the points raised during the review process.

We look forward to receiving your revised manuscript.

Kind regards,

Naranjargal Dashdorj

Academic Editor

PLOS ONE

Journal Requirements:

“NO”

“NO”

 This information should be included in your cover letter; we will change the online submission form on your behalf

Additional Editor Comments (if provided):

Dear authors,

This is an interesting paper. However, we would like you to improve the writing and address specific questions from the reviewers.

Reviewers' comments:

Reviewer's Responses to Questions

**Comments to the Author**

1. Is the manuscript technically sound, and do the data support the conclusions?

Reviewer #1: Yes

Reviewer #2: Partly

2. Has the statistical analysis been performed appropriately and rigorously? 

Reviewer #1: Yes

Reviewer #2: I Don't Know

3. Have the authors made all data underlying the findings in their manuscript fully available?

Reviewer #1: Yes

Reviewer #2: No

4. Is the manuscript presented in an intelligible fashion and written in standard English?

Reviewer #1: Yes

Reviewer #2: No

5. Review Comments to the Author

Reviewer #1: Need minor revision on sentence structure and syntax, specially introduction and discussion part.

Need more clarification on some findings noted in the review note.

Specify guidelines used in the examination and defining clinical activity score, and need correction on some definitions.

Reviewer #2: The authors present data from a cross-sectional study targeting correlations between TSH receptor stimulating antibody levels and Graves' orbitopathy

The focus of the study is a topic and worthy of interest and the sample size is appropriate, however the study in my opinion presents many critical elements and the conclusions of the authors do not appear to be supported by the data presented

In particular in both Groups GO and GD the duration of the disease was very high and the characteristics of orbitopathy depending on the natural history of the ocular complication can determine a number of confounding factors ; moreover, considering cases with long standing disease, how many of them were already treated for orbitopathy ?

In the presentation of the data the patients are divided regarding the patient's adherence to the thyrostatic treatment: this data cannot be an expression of the real thyroid compensation of the patient, who is known to greatly influence the orbitopathy, and therefore cannot replace the data on the real hormonal state of the patients .

In addition, from the reading of the demographic table it appears that out of 82 patients with GO only 5 do not do ATD . Of the 13 patients treated therefore with definitive surgical or radiometabolic therapy it seems to deduce that 8/13 patients have relapsed and therefore continue therapy with antithyroids ... What are the characteristics of these patients? Still among the 5 who do not take ATD how many take L-thyroxine? What is the thyroid status of these patients? Is there a correlation between hormonal control and the severity or activity of eye disease?

In reading the same table, in the GD group the authors state that 6 patients do not take antithyroids and 74 take them: so there seems to be an error in the overall number (80 patients vs 81 declared patients)

As already reported regarding the assimilation of the data on adherence to therapy as evidence of hormonal compensation, in the discussion of the paper, in my opinion the statement that the levels of TSI and TRAb correlated with the thyroid state, however plausible, is an apodictic statement.

-regarding the discrepancy between TSI levels in patients at risk for vision, although they claim that the figure is not significant due to the low number of cases, do not the authors believe that the tendentially lower levels of TSI partially contradict the proposed thesis?

It is also unclear to me the data regarding the percentage of hypertensive patients in GD compared to GO in terms of prevalence of hypertension, what is according to the authors the explanation of this data?

Minor comments

-I recommend amending reference 17 with the following reference to the latest eugogo guidelines

Bartalena, L., Kahaly, G. J., Baldeschi, L., Dayan, C.M., Eckstein, A., Marcocci, C., Marinò, M., Vaidya, B., Wiersinga, W.M., & EUGOGO † (2021). The Clinical Practice Guidelines of the European Group on Graves Orbitopathy (EUGOGO) 2021 for the medical management of Graves' orbitopathy. European Journal of Endocrinology, 185(4), G43–G67. https://doi.org/10.1530/EJE-21-0479

- In figure 2 " Grave's" is to be corrected in "Graves' "

6. PLOS authors have the option to publish the peer review history of their article (what does this mean?). If published, this will include your full peer review and any attached files.

Reviewer #1: No

Reviewer #2: No

---

## [Author Response · Author response to Decision Letter 0]

22 Sep 2022

Letter response to reviewers 

Dear reviewers, 

We would like to take this opportunity to express our thanks to the reviewers for the positive feedback and helpful comments for correction and modification.

We found the reviewers' comments to be helpful in revising the manuscript and have carefully considered and responded to each suggestion.

Below is our response to each point raised by the reviewers. 

We very much hope the revised manuscript is accepted for publication in PLOS ONE.

Response to Reviewers

Comments from Reviewer 1

Comment 1: Need minor revision on sentence structure and syntax, specially introduction and discussion part.

Response: The manuscripts' sentence structure and syntax were edited by a professional English editor, Springer Nature.

Comment 2: Need more clarification on some findings noted in the review note.

Response: I agree with the findings you noted in the review note and comments. The findings were revised and can be seen in “Revised Manuscript with Track Changes”.

• TRAB which was in the Introduction section, was replaced abbreviation TSHR-ab. Changes can be seen on the line 73.

TRAB: thyroid-stimulating hormone receptor autoantibodies (TSHR: Thyrotropin receptor)

TSHR-ab: thyroid-stimulating hormone receptor autoantibodies 

• Orbital involvement in Patients, Methods and Materials section, ocular symptoms added to be clear. Changes can be seen on the line 91.

• The moderately severe stage was corrected to moderate to severe is described on the line 103.

• There is no statistically significant difference between smoking and CAS score in GO patients, Although CAS score is higher in GO smoker patients. 

Comments 3: Specify guidelines used in the examination and defining clinical activity score, and need correction on some definitions.

Response: CAS is the most widely used classification of Graves’ ophthalmopathy. Based on this, a part of the EUGOGO category was created. Seven items to evaluate the GO patients have been clarified. Changes can be seen on the line 98.

Comments from Reviewer 2

Comment 1: The authors present data from a cross-sectional study targeting correlations between TSH receptor stimulating antibody levels and Graves' orbitopathy

The focus of the study is a topic and worthy of interest and the sample size is appropriate, however the study in my opinion presents many critical elements and the conclusions of the authors do not appear to be supported by the data presented.

In particular in both Groups GO and GD the duration of the disease was very high and the characteristics of orbitopathy depending on the natural history of the ocular complication can determine a number of confounding factors ; moreover, considering cases with long standing disease, how many of them were already treated for orbitopathy?

Response: The patients diagnosed with GO, did not have any systemic treatment and orbital surgery for orbitopathy. Twenty two of the patients use lubricant eye drops for symptoms of dry eye. Based on your comments, we have added this information to the method section in the manuscripts. Changes can be seen on the line 107.

Comment 2: In the presentation of the data the patients are divided regarding the patient's adherence to the thyrostatic treatment: this data cannot be an expression of the real thyroid compensation of the patient, who is known to greatly influence the orbitopathy, and therefore cannot replace the data on the real hormonal state of the patients.

In addition, from the reading of the demographic table it appears that out of 82 patients with GO only 5 do not do ATD . Of the 13 patients treated therefore with definitive surgical or radiometabolic therapy it seems to deduce that 8/13 patients have relapsed and therefore continue therapy with antithyroids ... What are the characteristics of these patients? Still among the 5 who do not take ATD how many take L-thyroxine? 

Response: The table 1 in manuscript shows that 77 of 82 patients with GO took ATD. 3 patients received radioiodine treatment: 2 were taking ATD and L-Thyroxine respectively and another one has no treatment. 10 patients underwent 2 different types of thyroidectomy: 3 were total thyroidectomy and 7 were subtotal thyroidectomy, which leaves a small unilateral or bilateral remnant in situ. 3 patients who had total thyroidectomy were taking L-thyroxine. All 7 patients who underwent thyroidectomy were treated with ATD. The information of L-Thyroxine and thyroidectomy are included. Changes can be seen in Table 1.

GO patients 82

ATD 77

L-Thyroxine 4

Thyroidectomy:

 Total thyroidectomy

 Subtotal thyroidectomy 10

3

7

Radioiodine:

 ATD

 L-Thyroxine

 No treatment 3

1

1

1

Commet 3: What is the thyroid status of these patients? 

Response: The purpose of our study was not to determine thyroid status. Therefore, we cannot provide information in this area. Thyroid status is always fluctuating. However, the old thyroid status of the patients was recorded. But their laboratory test date and laboratory method all were different. Therefore, the data on thyroid status could not be analyzed.

Comment 4: Is there a correlation between hormonal control and the severity or activity of eye disease?

Response: As mentioned above, previous thyroid status of patients were recorded. Even that, we cannot say whether there is a correlation with the severity or activity of eye disease. Other research studies show no correlations. 

Comment 5: In reading the same table, in the GD group the authors state that 6 patients do not take antithyroids and 74 take them: so there seems to be an error in the overall number (80 patients vs 81 declared patients)

Response: Table 1 shows that 74 of 81 patients with GD took ATD. Two patients received radioiodine treatment and treating with L-Thyroxine. Seven patients underwent thyroidectomy: 3 were total thyroidectomy and 4 were subtotal thyroidectomy.

Three patients who had total thyroidectomy were taking L-thyroxine and 2 patients who underwent thyroidectomy were treated with ATD. The information of L-Thyroxine and thyroidectomy are included. Changes can be seen in Table 1.

GD patients 81

ATD 74

L-Thyroxine 5

No treatment 2

Thyroidectomy:

 Total thyroidectomy

 Subtotal thyroidectomy 7

3

4

Radioiodine:

 L-Thyroxine 2

2

Comment 6. As already reported regarding the assimilation of the data on adherence to therapy as evidence of hormonal compensation, in the discussion of the paper, in my opinion the statement that the levels of TSI and TRAb correlated with the thyroid state, however plausible, is an apodictic statement.

-regarding the discrepancy between TSI levels in patients at risk for vision, although they claim that the figure is not significant due to the low number of cases, do not the authors believe that the tendentially lower levels of TSI partially contradict the proposed thesis?

Response: We were also surprised to see that the TSI values were lower in the sight-threatening stage. If we see this group, it included patients who had suffered from GO for many years and had poor treatment control. We hypothesize in two ways. First, In the mоderate to severe stage, the inflammatory process was very high. When the structural changes that have already occurred, the symptoms aggravate each other. However, Graves' disease itself has gone into remission, but once the structural changes have become more severe, it has progressed to sight-threatening stage. At the time of our study, TSI levels were reduced because the thyroid antibodies settled down. Another hypothesis is that the number of participants in the study is insufficient. We believe that the answer will be found by studying the sight-threatening stage in detail in the future.

Comment 7: It is also unclear to me the data regarding the percentage of hypertensive patients in GD compared to GO in terms of prevalence of hypertension, what is according to the authors the explanation of this data?

Response: We agree with you. Hypertension was greater in GD patients. But we did not focus on it. Further research is necessary.

Table 1 shows the mean age of the GO patients is 42.63 ±11.03, which is relatively younger than the GD patients 47.49 ±10.87. Нypertension increases with age.

---

## [Editor Report · Decision Letter 1]

19 Oct 2022

Association of clinical course with thyroid-stimulating immunoglobulin in Graves’ ophthalmopathy in Mongolians

PONE-D-22-03512R1

Dear Dr. Sarantuya Jav,

We’re pleased to inform you that your manuscript has been judged scientifically suitable for publication and will be formally accepted for publication once it meets all outstanding technical requirements.

Kind regards,

Naranjargal Dashdorj

Academic Editor

PLOS ONE

---

## [Editor Report · Acceptance letter]

24 Oct 2022

PONE-D-22-03512R1 

Association of clinical course with thyroid-stimulating immunoglobulin in Graves’ ophthalmopathy in Mongolians 

Dear Dr. Jav:

I'm pleased to inform you that your manuscript has been deemed suitable for publication in PLOS ONE. Congratulations! Your manuscript is now with our production department. 

Kind regards, 

on behalf of

Dr. Naranjargal Dashdorj 

Academic Editor

PLOS ONE